# Pancytopenia Due to Vitamin B12 and Folic Acid Deficiency—A Case Report

**Zoé Depuis [1],\*, Sophie Gatineau-Sailliant [1], Olivier Ketelslegers [2], Jean-Marc Minon [2], Marie-Christine Seghaye [1], Myriam Vasbien [2] and Marie-Françoise Dresse [1]**

[1] Department of Pediatrics, University Hospital Liège, Avenue de l'Hôpital 1, 4000 Liege, Belgium; s.gatineau@chuliege.be (S.G.-S.); mcseghaye@chuliege.be (M.-C.S.); mf.dresse@chuliege.be (M.-F.D.)

[2] Department of Laboratory Medicine, Regional Hospital of Liège, Boulevard du 12ème de Ligne 1, 4000 Liege, Belgium; olivier.ketelslegers@chrcitadelle.be (O.K.); jean.marc.minon@chrcitadelle.be (J.-M.M.); myriam.vasbien@chrcitadelle.be (M.V.)

\* Correspondence: zoe.depuis@chuliege.be

**Abstract:** We report a case of severe pancytopenia in a 15-year-old patient due to a severe deficiency in vitamin B12 and folic acid, probably of nutritional origin. The clinical and biological course was favorable after vitamin supplementation. With this case, we discuss the diagnostic approach of pancytopenia with megaloblastic anemia in children and adolescents, as well as the mechanisms involved in vitamin B12 and B9 deficiency. Hypovitaminosis B12 is known in its severe form but its diagnosis is often made difficult by insidious signs and symptoms. Traditional intramuscular replacement therapy has now proven to be effective orally. The clinical manifestations of folic acid deficiency are relatively similar to those of vitamin B12 deficiency, reflecting their intricate co-enzymatic functions. Its supplementation is administered orally.

**Keywords:** pancytopenia; megaloblastic anemia; vitamin B12 deficiency; folic acid



## 1. Introduction

The identification of pancytopenia requires a rigorous diagnostic approach adapted to the clinical situation. The neoplastic cause is the first to be mentioned, but multiple etiologies, including vitamin deficiency, may be responsible for this relatively common occurrence. The aim of this case report is to highlight the characteristics of megaloblastic anemia as well as the mechanisms involved in vitamin B12 and B9 deficiencies, which are ubiquitous coenzymes of essential pathways of intracellular metabolism.

## 2. Case Presentation

A 15-year-old female patient was referred to our emergency department by her attending physician based on suspicion of acute leukemia. She was initially consulted for asthenia and dyspnea on exertion that progressed for a month. Blood analysis showed a reduction in all three classes of formed elements with severe normochromic and aregenerative normocytic anemia, associated with moderate thrombocytopenia and leuko-neutropenia (Table 1).

The patient's medical history was marked by uncomplicated left renal agenesis. She had an episode of anorexia a year earlier with induced vomiting and a 15% weight loss and adopted a restrictive regime excluding meat products. There was no personal or family history of hemopathy or autoimmune disease. She had no abdominal pain or diarrhea. On admission, her hemodynamic state was stable. Clinical examination revealed moderate tachycardia, 2/6 systolic murmur at all auscultatory foci and mild conjunctival jaundice. There was no tumor syndrome or hemorrhagic sign. The additional blood test confirmed the pancytopenia accompanied by signs of hemolysis with lactate dehydrogenases (LDH) increased to 5200 U/L (standard: 126–259 U/L), a free bilirubin at 2 mg/dL (standard:

<1 mg/dL) and collapsed haptoglobin. The direct Coombs test was negative. The iron balance was normal, with serum iron at 87 μg/dL (standard: 30–160 μg/dL), ferritin at 208 μg/L (standard: 16 to 150 μg/L), transferrin at 1.94 g/L (standard: 2–4 g/L) and a normal transferrin iron saturation percentage of 32% (standard: 30–40%). C-reactive protein level was low at 0.7 mg/L (standard: <5 mg/L). On the blood smear, anisopoikilocytosis was visualized, as well as basophilic punctures and dacryocytes.

**Table 1.** Hemogram parameters.

| Parameters | Values | Normal Range |
|---|---|---|
| Hemoglobin | 4.3 | 10.9–15.3 g/dL |
| Mean corpuscular volume | 94 | 78–95 μm$^3$ |
| Mean corpuscular hemoglobin concentration | 32.6 | 31.7–35.4 g/dL |
| Mean corpuscular hemoglobin content | 30.4 | 25.4–32.7 pg |
| Reticulocyte count | 11.3 (0.8%) | 42–65 × 10$^3$/mm$^3$ |
| Platelet count | 100 | 194–345 × 10$^3$/mm$^3$ |
| Leucocyte count | 2.6 | 4.2–9.5 × 10$^3$/mm$^3$ |
| Absolute neutrophil count | 0.9 | 1.8–7.5 × 10$^3$/mm$^3$ |

A bone marrow puncture revealed rich cellularity with dysplasia of the erythroid (including megaloblastosis) and granulocytic (including gigantism) lineages, as well as of the megakaryocytic lineage to a lesser extent. There was no blast infiltration. The appearance was compatible with a deep vitamin deficiency (Figure 1). The cytogenetics and molecular biology returned normal.

Serum vitamin B12 was undetectable, lower than 100 ng/L (standard: >200 ng/L or 200 pg/mL). Serum folic acid (or vitamin B9) level was also below the limit of quantification, lower than 2 μg/L (standard: 3.9–26.8 μg/L). Total homocysteinemia was high at 90 μmol/L (standard: <15 μmol/L), consistent with this double vitamin deficiency.

In view of the clinical complaints and moderate hemodynamic tolerance, the patient received three transfusions of erythrocyte concentrates. Vitamin B12 supplementation was initiated with a daily intramuscular dose of 1 mg for three days, followed by an oral relay with one tablet of 1 mg per day. A folic acid substitution at a dosage of 5 mg per day for one week then 1 mg per day was also administered.

Biermer's anemia was ruled out by negative results for anti-intrinsic factor antibodies and antiparietal cell antibodies. Etiological assessment was completed by a search for *Helicobacter pylori*, a determination of fecal calprotectin, a thyroid assessment and a gastroscopy. These additional examinations were normal. The profile of urinary organic acids and plasma acylcarnitines was normal, and did not suggest a primary metabolic abnormality.

The diagnosis of dysmyelopoiesis with pancytopenia of deficient origin was retained. A dietetic support was established. Serum vitamin B12 normalized to 535 ng/L and serum folate to 13.9 μg/L, from the fifth day of vitamin supplementation. The patient's blood count normalized from the 10th day, with a reticulocyte crisis observed on the 6th day of treatment. The initial clinical signs improved. Supplementation was stopped at 6 weeks. The patient's clinical and biological course was favorable.

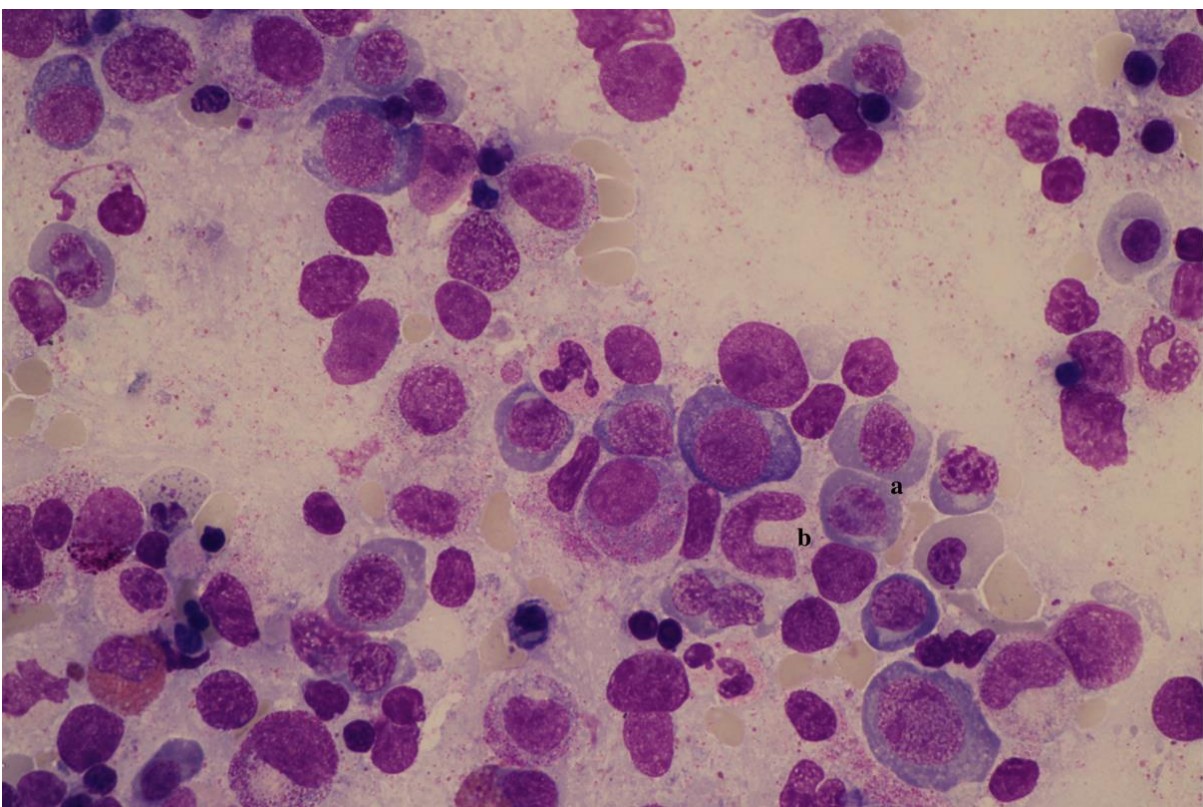

**Figure 1.** Myelogram. Erythroid hyperplasia with megaloblasts presenting asynchronism of nucleo-cytoplasmic maturation (a) and giant metamyelocyte (b).

## 3. Discussion

### 3.1. Pancytopenia and Megaloblastic Anemia in Children

Pancytopenia is defined as a reduction in the three cell lines of peripheral blood: erythrocytes, leukocytes and platelets [1]. This term corresponds to multiple definitions in the literature for children. We retain the association of anemia with a hemoglobin level lower than 2 standard deviations for age and sex, thrombocytopenia with a rate lower than $120 \times 10^3/mm^3$ and neutropenia with an absolute neutrophil count less than $1.5 \times 10^3/mm^3$ [2]. The discovery of pancytopenia first requires urgent evaluation of the hemodynamic tolerance of anemia, as well as determination of bleeding and infectious risk [1,2]. It is interesting to mention the relatively well-tolerated nature of anemia in our clinical case, in relation with the progressive onset of deficiency anemia.

In this patient, a first etiological orientation was suggested through characterization of the anemia, including analysis of the mean corpuscular volume (MCV), the mean corpuscular hemoglobin content and reticulocytosis. Vitamin deficiency anemia is most often macrocytic and aregenerative, illustrating its central origin. However, the MCV may be within normal values, as in the case of our patient. Hemolysis signs, usually not reported in a deficiency situation, may be disturbed with elevation of LDH and bilirubin due to ineffective erythropoiesis. The performance of an iron and hemolysis assessment as well as serum dosage of vitamins B12 and B9 is, therefore, an integral part of the initial assessment. The blood smear provides interesting although non-specific elements: anisopoikilocytosis with the presence of numerous macrocytes, dacryocytes and, in the case of vitamin B12 deficiency, schistocytes. Hypersegmented neutrophils are also observed [3].

Analysis of the myelogram is essential to determine the cellular richness and morphology of the elements. In the case of vitamin deficiency, bone marrow is often very rich, by excess of large erythroblasts (megaloblasts). The cell nucleus of these erythroblasts retains an immature appearance, while the cytoplasm continues to differentiate by enriching itself in hemoglobin (nucleocytoplasmic maturation asynchronism). The fragility of the

erythroblasts leads to apoptosis and intramedullary hemolysis, inducing compensatory hyperplasia. The granulocyte line shows gigantism (giant metamyelocytes) and numerous nuclear form abnormalities. Megakaryocytes can be reduced in number [3]. In addition to a cytological examination, a bone marrow karyotype, a molecular biology study and an immunophenotypic analysis are required if myelodysplasia or leukemia is suspected.

The causes of pancytopenia are numerous and differ in presentation and severity; their prevalence varies greatly from one country to another [4]. Bone marrow invasion by tumor cells (hematological or metastatic) must be quickly ruled out; the diagnosis of acute leukemia is the first suggestion when faced with a situation of pancytopenia in children [2,4,5]. Once malignant hemopathies have been ruled out, infectious causes (Epstein–Barr virus, cytomegalovirus, parvovirus B19, influenza, hepatitis virus, human immunodeficiency virus, bacterial sepsis) are the first to be described in developed countries, followed by non-tumor hematologic pathologies within the foreground acquired aplastic or constitutional anemia [4]. Macrophage activation syndrome and paroxysmal nocturnal hemoglobinuria should be mentioned in the differential diagnoses. Vitamin deficiency causes are poorly represented in children in developed countries such as in Europe or the United States, unlike in developing countries. In these, deficiency origins predominate, followed by infectious etiologies whose pathogens (salmonella typhi, leishmania, plasmodium) differ [5]. The iatrogenic causes must be sought; some antiepileptics and antibiotics, among others, can induce aplastic anemia [1,4]. Although rare, hereditary metabolic disorders are to be mentioned before pancytopenia without obvious cause [6,7]. An alteration of the three blood lines can be found in organic aciduria or lysosomal disease. Megaloblastic anemia associated with hyperhomocysteinemia and homocystinuria suggests an abnormal metabolism of vitamin B12 and folate [6].

### 3.2. General Information on Vitamin B12

Vitamin B12 or cobalamin is a water-soluble vitamin derived exclusively from the consumption of animal products, meat or dairy. Its daily intake varies between 5 and 7 µg, according to the diet. The recommended daily requirement is 0.8 µg up to the age of 3, with a progressive increase to 2.4 µg for adults [8]. Its storage is mainly hepatic and reserves are estimated between 2 and 5 mg; this corresponds to approximately three years of intake with a balanced diet. There is therefore a significant delay between an intake deficit and a cellular deficit, contributing to diagnostic difficulty [9,10]. Cobalamin is bound to food proteins. Under the action of pepsin and gastric fluid, it loosens and binds to haptocorrin, a transporter protein secreted by the salivary glands and gastric cells. This complex is lysed at the duodenal level by the action of pancreatic and biliary secretions. Vitamin B12 then binds with intrinsic factor (FI). Secreted by gastric cells, FI helps protect cobalamin from bacterial catabolism and binds to terminal ileal cells. At the same time, simple diffusion allows the absorption of 1 to 5% of the dose of cobalamin ingested [9].

The prevalence of vitamin B12 deficiency in children is difficult to estimate but has been increasing in recent years. It seems more important in disadvantaged socioeconomic backgrounds, in relation to a less balanced diet [11,12].

### 3.3. Etiologies of Vitamin B12 Deficiency

The etiologies of cobalamin deficiency arise from the physiology of its absorption. Insufficient intake, impaired exogenous gastric or pancreatic function as well as any damage to the integrity of the ileal mucosa are likely to result in a deficiency. The main causes are summarized in Figure 2. Biermer's anemia or pernicious anemia, a consequence of an autoimmune-induced FI deficiency, represents only a small proportion of these etiologies. The same is true for nutritional deficiency. Maldigestion, or syndrome of non-dissociation of vitamin B12 from its carrier proteins, food or haptocorrins, is the most common cause [13]. It includes pathologies altering the transport system and, therefore, its absorption, such as infection with *Helicobacter pylori*, atrophic gastritis, pancreatic insufficiency, bacterial overgrowth or even the intake of metformin or proton pump inhibitors [13]. In infants, the

main etiologies are maternal vitamin B12 deficiency, hereditary transcobalamin deficiency, congenital FI deficiency and congenital malabsorption [3].

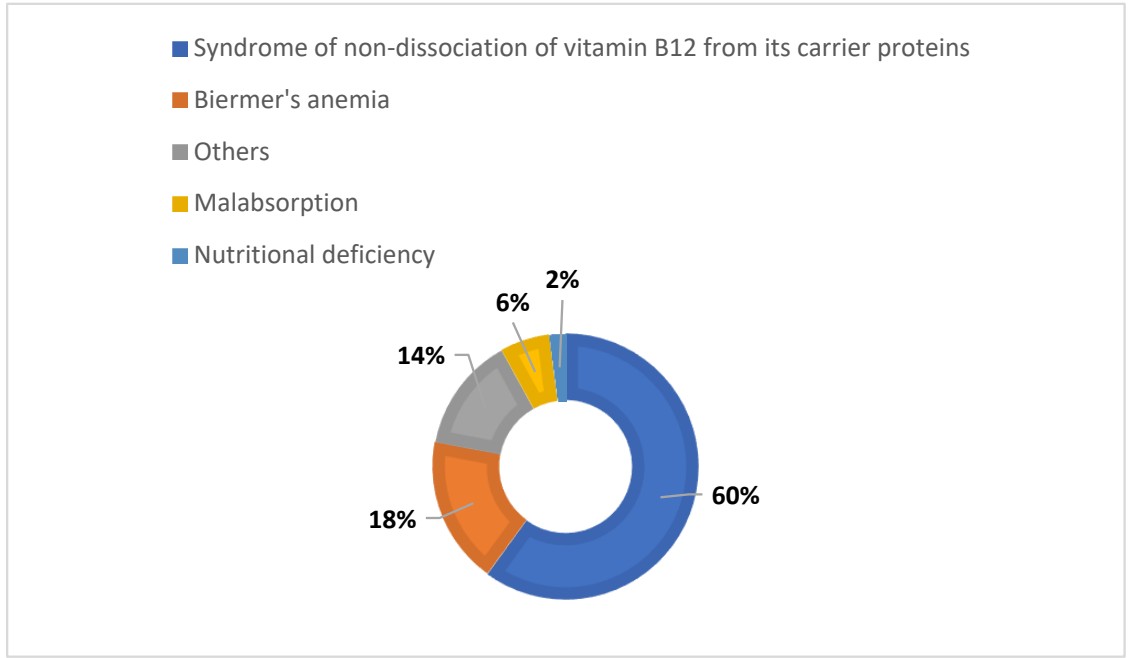

**Figure 2.** Graphical presentation of main etiologies of vitamin B12 deficiency, according to Rufenacht et al. [9] and Bosco et al. [10].

### 3.4. Biological Markers of Vitamin B12 Deficiency

There are three serum B12 transporters, of which holotranscobalamin (holoTC II) appears to be the most active. The serum vitamin B12 dosage reflects the amount of cobalamin bound to all transporters. The dosage of HoloTC II is said to be more precise but less widespread, and its homeostasis is still not understood [9,14,15]. The management of hepatic reserves, the existence of a renal reabsorption system and an enterohepatic cycle modulate this serum vitamin B12 level.

Cobalamin is a ubiquitous coenzyme involved in two pathways of intracellular metabolism. Firstly, it participates in the transformation of methylmalonic acid (MA) into succinic acid, a metabolite of the citric acid cycle. Then, it promotes the conversion of homocysteine (HC) to methionine and allows use of folate in the synthesis of purines. HC and MA are therefore markers of these metabolic pathways; their increase reflects a cellular cobalamin deficiency [15].

Recent studies are based on the total serum vitamin B12 level, which should be less than 200 pg/mL in two separate dosages, or on the combination of a serum vitamin B12 level below 200 pg/mL and elevation of MA or HC to diagnose a deficiency [9,14]. Ideally, the diagnosis of B12 deficiency should be made on the basis of the concordant assay of two of these three biomarkers, each having limitations [15].

### 3.5. Signs and Symptoms of Vitamin B12 Deficiency

As Vitamin B12 is involved in the formation of DNA, its deficiency is preferentially reflected through rapidly replicating cells such as hematopoietic, skin and digestive cells. Understanding of the myelin damage that causes neurological manifestations remains incomplete. The clinical expression of cobalamin deficiency is highly variable (Table 2) [16].

**Table 2.** Main clinical and biological manifestations of vitamin B12 deficiency.

| | |
|---|---|
| Hematological | • Macrocytosis<br>• Aregenerative macrocytic anemia<br>• Medullary megaloblastosis<br>• Hypersegmentation of neutrophils<br>• Elevation of bilirubin and LDH<br>• Decrease in haptoglobin<br>• Thrombocytopenia and neutropenia<br>• Pancytopenia<br>• Thrombotic pseudomicroangiopathy (schistocytes) |
| Digestive and cutaneous | • Jaundice<br>• Atrophic glossitis<br>• Recurrent mucocutaneous ulcerations<br>• Hyperpigmentation |
| Neurological and psychiatric | • Peripheral neuropathy<br>• Ataxia, cerebellar syndrome<br>• Combined sclerosis of the spinal cord<br>• Damage to the cranial nerves (optic neuritis)<br>• Urinary and/or fecal incontinence<br>• Impairment of higher cognitive functions<br>• Depression, psychosis |

In the longer term, it is interesting to note that hyperhomocysteinemia increases the risk of atherosclerosis. Studies in children and adolescents have shown that severe cobalamin deficiency is associated with early endothelial dysfunction. Vitamin supplementation appears to reduce cardiovascular risk [12,17].

*3.6. Therapeutic Strategy*

The aim of the treatment is to overcome cellular deficits and restore the vitamin stock. Vitamin B12 substitution is conventionally carried out through intramuscular injection of cyanocobalamin. Different administration regimens can be found in the literature. In adults, the oral route has shown to be effective through passive absorption at the ileal level without FI intervention [9,10,14].

Indeed, various studies have reported that oral supplementation with high doses of vitamin B12 (1000 to 2000 mcg initially daily, thereafter weekly and then monthly) may be as effective as intramuscular administration in deficient adults [18]. In children from 6 months of age, the effectiveness of oral vitamin substitution has recently been documented for nutritional deficiencies. Oral treatment could, therefore, be offered as a first-line treatment, with adequate compliance [19]. It presents various advantages, including being more suitable for a pediatric population with a lower cost. The bioavailability of vitamin B12 by passive diffusion is around 1%. Relative to the interpersonal variability of absorption, the dosage should be adjusted with a minimum daily dose of 0.5 to 1 mg. In case of doubt about patient compliance, neurological disorders or impaired ileal integrity, the parenteral route remains indicated [19,20].

The duration of substitution varies depending on the etiology. In a reversible cause, the resolution of clinical signs as well as the normalization of serum cobalamin levels and blood count, generally requiring at least one month of treatment, would be criteria for deciding to discontinue treatment [20]. For patients with anemia, a hematological response with a reticulocyte crisis should be seen within 7–10 days, provided their folate and iron

levels are correct. Indeed, another masked previous deficiency or co-existing etiology should be considered in case of suboptimal response [21].

### 3.7. Folic Acid Deficiency

Vitamin B9 or folic acid is a water-soluble vitamin whose contributions are mainly food (green vegetables, fruits, egg yolks, meats). The recommended daily requirement for infants is 70 µg. For children until puberty, requirements are between 100 and 300 µg, then between 200 and 400 µg beyond [8,22].

The main cause of deficiency is insufficient food intake. Folate is absorbed from the proximal small intestine. Almost half of the body's folate is located in the liver. The bioavailability of natural folates depends on many factors related to their cooking process and to the intestinal lumen; these factors have less influence on folic acid used in supplements, fortified foods or treatments, which will therefore have better bioavailability. A serum folate level lower than 3 µg/L (or 7 nmol/L) indicates a folate deficiency. In most cases, this rate is sufficient and routine red cell folate testing seems not necessary. Interpretation of the serum folate level according to the clinical picture is important [21]. Cases of isolated clinical folate deficiency in children and adolescents are rare in developed countries, and its exact prevalence seems difficult to estimate. In practice, a condition leading to a deficit or malabsorption of several nutrients should be investigated.

The roles of vitamins B9 and B12 are strongly intertwined. The clinical manifestations of folate deficiency, reflecting the failure of its coenzyme function, are relatively similar to those of vitamin B12 deficiency, of varying presentation and severity.

Replacement therapy is orally administered with dosage of 1 to 5 mg per day, depending on the etiology and severity of clinical presentation. It should be noted that supplementation with folic acid may initially mask a vitamin B12 deficiency and, thus, delay its management, at the risk of progression of neurological damage, some consequences of which will be irreversible [23]. Care must therefore be taken not to ignore a B12 deficiency before replacement treatment with folic acid, in order to add supplementation if necessary.

### 3.8. Vitamin Deficiency among Vegetarians and Vegans

Various studies have documented that vegetarians, and particularly vegans, have lower levels of vitamin B12 than omnivores, related to the fact that intakes come mainly from the consumption of animal products. Higher deficiency prevalence was reported among vegetarians and vegans [24]. However, they seem to have higher folate concentrations compared to omnivores [25].

Therefore, cobalamin status should be monitored for individuals following this type of diet, as they are at higher risk. Oral supplementation or fortified foods may be considered according to the clinical situation. This supplementation is particularly recommended for vegans and during pregnancy and breastfeeding, which are periods at risk of deficiency for both the pregnant woman and her fetus or infant [21].

## 4. Conclusions

The diagnosis of pancytopenia with megaloblastic anemia due to vitamin deficiency is based on the concordance of anamnestic, clinical, cytological elements and the assay of associated biomarkers. The serum dosage of vitamins B12 and B9 should be included in the initial etiological assessment of pancytopenia, due to non-specificity of clinical signs. The manifestations of cobalamin and folate deficiency are relatively similar, reflecting their intricate co-enzymatic functions. In industrialized countries, provided the diet is balanced and in the absence of underlying disease, vitamin requirements are covered and deficiencies of nutritional origin are rare. Vitamin B12 substitution, traditionally administered intramuscularly, has been shown to be effective orally in children for nutritional deficiencies. Acid folic supplementation is administered orally. The importance of early diagnosis and prompt vitamin supplementation, due to potential complications, must be underlined.

**Author Contributions:** Conceptualization, Z.D. and S.G.-S.; validation, S.G.-S. and M.-F.D.; investigation, Z.D.; resources, Z.D. and S.G.-S.; writing—original draft preparation, Z.D. and S.G.-S.; writing—review and editing, O.K., M.V. and J.-M.M.; visualization, M.-F.D.; supervision, M.-F.D. and M.-C.S. All authors have read and agreed to the published version of the manuscript.

**Funding:** This research received no external funding.

**Institutional Review Board Statement:** Not applicable.

**Informed Consent Statement:** Informed consent was obtained from all subjects involved in the study. In addition, information is completely anonymized and will not adversely affect the rights of the patient.

**Data Availability Statement:** Not applicable.

**Conflicts of Interest:** The authors declare no conflict of interest.

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
