# Peer review of "Pancytopenia Due to Vitamin B12 and Folic Acid Deficiency—A Case Report"

_pediatrrep, doi:10.3390/pediatric14010016_

Round 1

Reviewer 1 Report

I read the paper titled “Pancytopenia due to Vitamin B12 and Folic Acid Deficiency – A Case Report

” from Depuis et al. The manuscript is very well-written and I have no real concerns at all with it. I have only the suggestion to provide more bibliographical references. I believe a greater number of bibliographic references allows the opportunity to deepen the topic.

For example authors can give references where they write about therapy.

I also suggest to give some informations about the risk of exclusion diets and if there is the possible to counter these risks taking vitamin B12 as supplements

Best regards 

Author Response

Dear Reviewer,

Many thanks for your very helpful and relevant review and comments. 

Point 1: As you suggested, we have added six additional bibliographical references to the manuscript. These interesting articles provide additional information on the subject, especially from a treatment perspective.  

Point 2: Also as you suggested, we have added a short paragraph at the end of the article regarding the risks of a restrictive, vegetarian or vegan diet. Indeed, higher vitamin B12 deficiency prevalence was reported among vegetarians and vegans, and it was important to mention this in the clinical context of our case report. Cobalamin status should be monitored for individuals following this type of diet, as they are at higher risk. Oral supplementation or fortified foods may be considered to counter this risk, according to the clinical situation. This supplementation is particularly recommended for vegans and during pregnancy or breastfeeding. Thank you for your suggestion to mention it.

Best regards,

Zoé Depuis

Reviewer 2 Report

In the first sentence of the 14th paragraph the authors should specify whether there is a different daily intake of Vitamin B12 between adults and children. 

In the fifth line of the sixteenth paragraph and in the fifth line of the twentieth paragraph the authorsm should replace the acronym IF with FI (as specified in the tenth line of the fifteenth paragraph)

Author Response

Dear Reviewer,

Many thanks for your review and your relevant comments.

As you suggested, there is indeed a difference in the daily intake of vitamin B12 in children and adults. We have therefore made this clear in the discussion. The recommended daily requirement for vitamin B12 is 0.8 µg up to the age of 3, with a progressive increase to 2.4 µg for adults. 

Then, we have replaced the acronym IF with FI in the both locations concerned.

Thanks again for your suggestions.

Best regards,

Zoé Depuis

Reviewer 3 Report

This is an interesting and well-written case report with a mini-review about pancytopenia linked to vitamin b12 and/or b9 deficiencies in children.

I have one major comment:

I can't find the serum vitamin B12 and vitamin B9 values in your manuscript, even in the paragraph starting at line 71. These values must be precised in this manuscript, together with the standard of your laboratory.

And I have minor comments :

Line 59: can you precise the value of the iron saturation of transferrin? (32% ?)

Line 59: what marker(s) did you use to say there is no inflammatory syndrome? (CRP ? if so, can you precise the value). I'm not sure leucocytes are enough as you have a pancytopenia, and ferritin is not low.

Line 132-133: I would change with something like: “… reprensented in children in Europe, unlike in developing countries.”

Figure 2: I don’t really understand from where come the values in Figure 2, except that we can find them in the figure 2 of your reference 9 (rev med Suisse). If you used this reference only to provide these values, you should add in the legend of your figure something like “according to Bosco et al” of “adapted from…”

The paragraph Folic acid deficiency is quite short in comparison with the previous ones. This is not really a problem as this deficiency is probably less frequent. But in my opinion you shoud add the normal or threshold value to diagnose folic acid deficiency and the prevalence (0,3% ?). I am not sure the etiologies are really the same, I would attenuate more the word similar, maybe “relatively” or “quite similar”.

Author Response

Dear Reviewer,

Many thanks for your review and your very helpful and relevant comments.

Concerning your major comment, you are absolutely right and these values needed to be specified, their deficit being the subject of this case report. We have now specified them in the case description, together with the standard of our laboratory. Serum vitamin B12 was indosable for our patient, lower than 100 ng/L (standard: > 200 ng / L or 200 pg / mL). Serum folic acid (or vitamin B9) level was also below the limit of quantification, lower than 2 µg / L (standard: 3,9 – 26,8 µg / L). We also specified the normalized values after 5 days of supplementation : vitamin B12 normalizes to 535 ng / L and serum folate to 13,9 µg / L.

Concerning your minor comments :

Point 1. The transferrin iron saturation percentage of the patient was 32% (standard: 30-40%) and we have now clarified this in the case description.

Point 2. As you suggested, we precised the value of C-reactive protein, that was low at 0.7 mg/L (standard: < 5 mg / L). Indeed, the leukocyte count was not interpretable in this context to judge the absence of an inflammatory syndrome.

Point 3. We have changed to clarify that these considerations relate to children in developed countries such as those in Europe and the United States.

Point 4. In fact, these values are taken from two of the articles from Rev med Suisse (references 9 and 10 in our revised paper) and as you rightly suggested, we have mentioned them in the legend.

Point 5. The paragraph on folic acid deficiency is indeed relatively short; our intention was to develop the subject mainly from the point of view of pancytopenia with megaloblactic anemia and vitamin B12 deficiency. However, as you rightly suggest, some additional information allows a better understanding, as well as enriching the article. We have therefore developed the paragraph a little further. A serum folate level lower than 3 µg / L indicates a folate deficiency. In most cases, this rate is sufficient and routine red cell folate testing seems not necessary. Cases of isolated clinical folate deficiency in children and adolescents are rare in developed countries, and its exact prevalence seems really difficult to estimate. A condition leading to a deficit or malabsorption of several nutrients should be investigated when folate deficiency is discovered.

We have also enhanced the conclusion with the above considerations.

Thank you so much for your suggestions.

Best regards,

Zoé Depuis